# The Effectiveness of Mass Transfer in the MHD Upper-Convected Maxwell Fluid Flow on a Stretched Porous Sheet near Stagnation Point: A Numerical Investigation

**Anwar Shahid** 

College of Astronautics, Nanjing University of Aeronautics and Astronautics, Nanjing 210016, China;
anwar@nuaa.edu.cn

**Abstract:** The present inquiry studies the influence of mass transfer in magnetohydrodynamics (MHD) upper-convected Maxwell (UCM) fluid flow on a stretchable, porous subsurface. The governing partial differential equations for the flow problem are reformed to ordinary differential equations through similarity transformations. The numerical outcomes for the arising non-linear boundary value problem are determined by implementing the successive linearization method (SLM) via Matlab software. The accuracy of the SLM is confirmed through known methods, and convergence analysis is also presented. The graphical behavior for all the parametric quantities in the governing equations across the velocity and concentration magnitudes, as well as the skin friction and Sherwood number, is presented and debated in detail. A comparability inquiry of the novel proposed technique, along with the preceding explored literature, is also provided. It is expected that the current achieved results will furnish fruitful knowledge in industrious utilities and correlate with the prevailing literature.

**Keywords:** Maxwell fluid; MHD; mass transfer; stretching sheet; porous media; SLM technique

## 1. Introduction

The earth demonstrates various exemplars of flows, regarding non-Newtonian fluids. Recently, the study of such fluids has been captivating researchers, being extensively studied throughout the last two decades. Indubitably, the developing governing equations for non-Newtonian fluids are strongly non-linear, high-order, and often more complicated than the Navier–Stokes equations. The flow of non-Newtonian fluids has importance in an expanded category of utilities, being involved in processing in industries including synthetic fibers, the squeezing of melted plastics, oil, gas well drilling, windup processes, and in certain flows of polymer solutions. An immense category of liquids and commercial usages has motivated investigators to analyze non-Newtonian fluid conduct. Non-Newtonian fluids have dissimilar features from Newtonian fluids. To understand comprehensively non-Newtonian fluids and their usages, it is essential to investigate their flow conduct. Researchers have performed analyses on the modeling of second-grade and third-grade fluids, which could not foretell the influence of stress relaxation. The Maxwell model, which is a sub-categorization of the rate-type fluid, has gained preference. Tan et al. [1] analyzed the unsteady, viscoelastic fluid flow of a fractional Maxwell model through parallel plates. Fetecau and Fetecau [2] performed a study in order to attain exact solutions for the Maxwell fluid flow through an infinite surface. Mokhopadhyay et al. [3] elaborated on the transpiration impact into unsteady magnetohydrodynamics (MHD) flow for an upper-convected Maxwell (UCM) fluid on a stretchable subsurface with a chemical reaction. The MHD flow and heat transferal for a UCM fluid past a stretched plate with varying thermophysical

characteristics were presented by Prasad et al. [4]. Anwar et al. [5] recorded the impact of the ramped wall temperature and ramped wall velocity for an unsteady MHD convective Maxwell fluid flow. The recent inquiries demonstrating the flows of Maxwell fluid can be found in [6–10].

The MHD fluid flows over a permeable medium, acting as a momentous function in agricultural and mechanical technologies, squeezing out petrol from fossil oils in petrol industries. The MHD tools have discovered extensive utilities in material sciences, and also in biomedical sciences. The usage of MHD microfluid tools has been massively utilized in numerous grasslands. The tri-biological efficiency of MHD nanofluids was discussed by Andablo-Reyes et al. [11]. Ellahi [12] scrutinized the non-Newtonian MHD nanofluid flowing through a pipe and recorded that the MHD parametric quantity decelerated the movement of fluid particles. For a varying viscosity, the velocity distribution was bigger, compared with the temperature distribution. Bhatti et al. [13] inspected the influence of varying magnetic fields in the peristaltic flow on a Jeffery fluid through a non-uniform rectangular duct. The impact of joule heating on the MHD flow of an upper-convected Maxwell fluid was elucidated by Zaidi and Mohyud-Din [14]. Furthermore, Hassan et al. [15] considered the impact of an oscillating magnetic field on ferrofluid. Mahabaleshwar et al. [16] investigated the effects of radiation and the Navier slip boundary on Walters' liquid B flow over a stretching sheet and through a porous medium. The systematic solutions for the time-dependent average velocity in the MHD peristaltic rotated flow of a couple stress fluids through a uniformly elastic pervious channel were analyzed by Krishna et al. [17]. Makinde et al. [18] explored the reactive MHD variant viscosity while flowing on a convectively heated subsurface from porous media, alongside radiative heat transfer and thermophoresis. Shah et al. [19] explored the numerical simulation of entropy optimization and the thermal conduct of a nanofluid through a permeable medium. Khan et al. [20] recorded MHD nanofluid flow, along with non-linear radiation, through a non-linear stretching and shrinking wedge. A few relevant studies on MHD non-Newtonian fluids flow can be found in [21–23].

The phenomenon of mass transportation in a system is the shifting of mass from one locale to another locale. This phenomenon has been utilized scientifically in diverse fields of science for assorted structures and systems, associating the molecular and convective transmission of molecules and atoms. Part of the ordinary specimen of mass transport procedures includes the vaporization of liquids, diffusing of chemical impurities into oceans and river-systems through artificially or naturally occurring reservoirs, and the segregation of compounds in the purification process. Mass transfer has expanded usage in industries and chemical engineering, too. In general, the transport of chemical species occurs via diffusion in a state, or via an interface within phases. The pushing efficacy of mass transferal is due to concentration variance, or the random movement of molecules, generating a net transfer of mass from one locality of a higher concentration to another locality of a lower concentration. The rate of mass transfer is computed via computations and coefficients of the mass transfer's utilities. Several researchers have exposed many more concerns about mass transfer when subjected to various features. For an instance, Liu [24] and Cortell [25] inquired into the hydromagnetic flow past a stretched subsurface, along with heat and mass transfer. The problem of a second-grade fluid saturating a permeable medium was examined by Akyildiz et al. [26]. Layek et al. [27] inspected the heat and mass transferal analysis of boundary value flow in a heated, stretched, permeable plate with heat absorption. Makinde [28] explored the combination of mixed convection, thermal radiation, and a chemical reaction through a vertically pervious subsurface. The viscoelastic flow and species transfer into a Darcian high-porous channel was studied by Anwer and Makinde [29]. Abbas et al. [30] examined the peristaltic propulsion for a Jeffrey nanofluid, along with the thermal radiation and chemical reaction influence. Khan et al. [31] elaborated on the combination of heat and mass transport for third-grade nanofluids on a convective heating, stretchable, pervious plate. Deebani et al. [32] presented the hall effect on radiative Casson fluid flow with a chemical reaction over a rotating cone through entropy optimization. Al-Khaled and Khan [33] inspected the thermal features of a Casson nanofluid, comprising microorganisms along with the temperature-dependent viscosity and variant

thermal conductivity. A few relevant inquiries on the transfer of mass and fluid flow through stretched surfaces can be found in [34–36].

Keeping in mind the above discussion, the objective of the present study for the Maxwell fluid model is to identify non-Newtonian fluid behaviors and analyze the impact of mass transfer in the MHD upper-convected Maxwell (UCM) fluid flow in the vicinity of the stagnation point past a stretched, permeable subsurface. The analysis of mass transfer past permeable surfaces has considerable significance as a result of its vast utilities. The preceding investigations were based on the continual physical aspects of the fluid. The current flow problem is reduced to non-linear ordinary differential equations through appropriate similarity transformations and resolved by employing the successive linearization method (SLM), providing purposeful results numerically [37–42]. The current technique proved to be very efficient and has a fast convergence rate in comparison with various numerical schemes. The impact of governing physical parametric quantities of concern is shown through graph curves and explicitly discussed.

## 2. Mathematical Formulation

Consider the incompressible and steady stagnation point flow of an upper-convected Maxwell (UCM) fluid constricted by a porous, stretched sheet on $\overline{y} = 0$. The flow engrosses the range $\overline{y} > 0$. The $\overline{x}$ and $\overline{y}$ axes are assumed to be alongside and perpendicular to the sheet, respectively, as shown in Figure 1. An external magnetic field $B_0$ is exerted in a transverse direction to the flow, and the electric and magnetic field is desolated due to the minimal magnetic Reynolds number. Additionally, the mass transfer factors are esteemed. The mass transferal is the flow incorporating the species $A$ that is barely solvable in fluid $B$. The concentration over the sheet subsurface and the solubility of $A$ into $B$ is $\tilde{C}_w$, the part traveling through the sheet is $\tilde{C}_\infty$, and the reaction rate is $k_1$. The velocity magnitudes of the stagnation point on $\overline{x} = 0$, $\overline{y} = 0$ are expressed as

$$\tilde{u}_e(\overline{x}) = a\overline{x}, \ \tilde{u}_e(\overline{y}) = a\overline{y}, \tag{1}$$

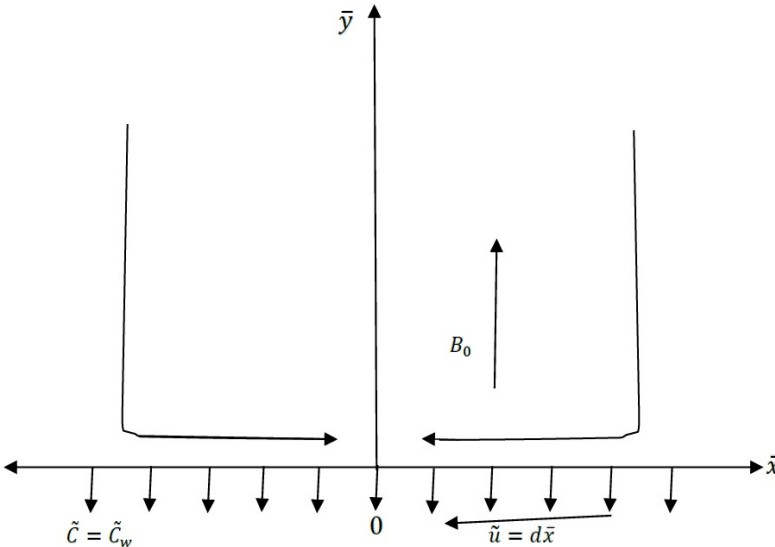

**Figure 1.** Geometry of the flow problem and coordinate system.

Here, the constant $a > 0$ is proportional to the free stream velocity parting through the stretched sheet. Through boundary layer approximations [43–45], the consequent flow equations are

$$\frac{\partial \tilde{u}}{\partial \overline{x}} + \frac{\partial \tilde{v}}{\partial \overline{y}} = 0 \tag{2}$$

$$\tilde{u}\frac{\partial \tilde{u}}{\partial \overline{x}} + \tilde{v}\frac{\partial \tilde{v}}{\partial \overline{y}} + \lambda_1 \left\{ \tilde{u}^2 \frac{\partial^2 \tilde{u}}{\partial \overline{x}^2} + 2\tilde{u}\tilde{v}\frac{\partial^2 \tilde{u}}{\partial \overline{x}\partial \overline{y}} + \tilde{v}^2 \frac{\partial^2 \tilde{u}}{\partial \overline{y}^2} \right\}$$
$$= v\frac{\partial^2 \tilde{u}}{\partial \overline{y}^2} + \tilde{u}_e \frac{d\tilde{u}_e}{d\overline{x}} - \frac{B_0^2 \sigma}{\rho}\left[ \tilde{u} - \tilde{u}_e + \lambda_1 \tilde{v}\frac{\partial \tilde{u}}{\partial \overline{y}} \right] - \frac{\mu_f}{k}\tilde{u} \tag{3}$$

$$\tilde{u}\frac{\partial \tilde{C}}{\partial \overline{x}} + \tilde{v}\frac{\partial \tilde{C}}{\partial \overline{y}} = \overline{D}_B \frac{\partial^2 \tilde{C}}{\partial y^2} - k_1\left( \tilde{C} - \tilde{C}_\infty \right) \tag{4}$$

Their boundary conditions are

$$\tilde{u} = \tilde{u}_w(\overline{x}) = d\overline{x}, \ \tilde{v} = -\tilde{v}_w, \ \tilde{C} = \tilde{C}_w \text{ on } \overline{y} = 0 \tag{5}$$

$$\tilde{u} = a\overline{x}, \ \tilde{C} = \tilde{C}_\infty \text{ at } \overline{y} \to \infty \tag{6}$$

where $\tilde{u}$ and $\tilde{v}$ are the velocity constituents alongside the $\overline{x}$- and $\overline{y}$-axis, $v$ is the kinematic viscosity, $\rho$ is the density, $\sigma$ is the electrical conductivity of the fluid, the relaxation time is $\lambda_1$, the mass diffusion is $\overline{D}_B$, the concentration field is $\tilde{C}$, $d$ is the stretching rate, and the reaction rate is $k_1$. It has been mentioned previously [44–47] that the extra expression $\frac{B_0^2 \sigma}{\rho}\left[ -\tilde{u}_e + \lambda_1 \tilde{v}\frac{\partial \tilde{u}}{\partial \overline{y}} \right]$ is in the momentum equation. This investigation embodies similar deduction to the MHD two-phase Maxwell fluid flow in the flourishing explorations.

The following expression was introduced in [45,46]:

$$\tilde{u} = d\overline{x}g'(\eta), \ \tilde{v} = -\sqrt{dv}\,g(\eta), \ \eta = \sqrt{\frac{d}{v}}\,\overline{y}, \ \phi(\eta) = \frac{\tilde{C} - \tilde{C}_\infty}{\tilde{C}_w - \tilde{C}_\infty} \tag{7}$$

where $a$ and $d$ are positive constants with dimensional reciprocals of time.

Equation (2) satisfies them equally, and Equations (3)–(6) arrive at

$$g''' + \{M\beta_1 + 1\}gg'' - g'^2 + \beta_1\left\{2gg'g'' - g^2g'''\right\} - M(g' - \alpha) - Kg' + \alpha^2 = 0 \tag{8}$$

$$\phi'' + S_c g\phi' - S_c K_c \phi = 0 \tag{9}$$

The associated boundary conditions are

$$g'(\eta) = 1, \ g(\eta) = S, \ \phi(\eta) = 1, \text{ on } \eta = 0 \tag{10}$$

$$g'(\eta) = \alpha, \ \phi(\eta) = 0, \text{ as } \eta \to \infty \tag{11}$$

where $S = \frac{-\overline{v}_0}{\sqrt{av}}$, $K = \frac{\mu_f}{kd}$, $M = \frac{\sigma \overline{B}_0^2}{\rho d}$, $\beta_1 = \lambda_1 d$, $\alpha = \frac{a}{d}$, $S_c = \frac{v}{D_B}$ and $K_c = \frac{k_1}{d}$ are the suction, porosity parameter, magnetic parametric quantity, Deborah number, Schmidt number, and chemical reaction parametric quantity, respectively. Further, $K_c > 0$ or $K_c < 0$ stands for destructive or generative chemical reactions, whereas $K_c = 0$ stands for non-reactive species.

**Physical Quantities**

The surface skin friction coefficient $C_g$ and local Sherwood number $Sh_{\overline{x}}$ are described as

$$C_g = \frac{2\tau_w}{\rho \tilde{u}_w^2}, \ Sh_{\overline{x}} = \frac{\overline{x}j_w}{D_B\left( \tilde{C}_w - \tilde{C}_\infty \right)} \tag{12}$$

where $\tau_w$ is the wall shear stress and $m_w$ is the mass flux, with solutions for both presented below:

$$\tau_w = \mu\left( \frac{\partial \tilde{u}}{\partial \overline{y}} \right)_{\overline{y}=0}, \ m_w = -D_B\left( \frac{\partial \tilde{C}}{\partial \overline{y}} \right)_{\overline{y}=0} \tag{13}$$

Placing Equation (13) into Equation (12) yields

$$\frac{1}{2}C_g\sqrt{Re_{\bar{x}}} = g''(0), \quad \frac{Sh_{\bar{x}}}{\sqrt{Re_{\bar{x}}}} = -\phi'(0) \tag{14}$$

where $Re_{\bar{x}}$ is the local Reynolds number.

## 3. Numerical Technique

We implemented the successive linearization method (SLM), assuming expansion [37–42], which is given as

$$g(\eta) = g_i(\eta) + \sum_{n=0}^{i-1} g_n(\eta), \quad (i = 1, 2, 3, \ldots) \tag{15}$$

where $g_i$ is the undetermined function, which is to be achieved iteratively. Presuming the earliest hypothesis $g_0$ of the mode, then

$$g_0 = S + \alpha\eta + \{1 - \alpha\}(1 - e^{-\eta}) \tag{16}$$

We write Equation (8) as

$$L = g''' - M(g' - \alpha) - Kg' \tag{17}$$

In addition, we can say

$$N = \{M\beta_1 + 1\}gg'' + \beta_1\{2gg'g'' - g^2g'''\} - (g')^2 + \alpha^2 \tag{18}$$

where L and N are the linear and non-linear segments, respectively. By inserting Equation (15) into Equation (8) and conceding the linear segment, we arrive at

$$g_i''' + C_{0,i-1}g_i''' + C_{1,i-1}g_i'' + C_{2,i-1}g_i' - [M + K]g_i' + C_{3,i-1}g_i + M\alpha = r_{i-1} \tag{19}$$

The respective boundary conditions become

$$g_i(0) = 0 = g_i'(0) = g_i'(\infty) \tag{20}$$

Equation (19) is solved through the Chebyshev spectral collocation technique. The conversion of a physical part toward a finite part $[-1, 1]$ is brought by applying a transformation of the following form:

$$\Gamma = \frac{2\eta - 1}{\Theta} \tag{21}$$

The $[-1, 1]$ is discretized. To make the nodal points into $[-1, 1]$, the Gause–Lobatto collocation is utilized:

$$\Gamma_I = \cos\frac{\pi i}{N}, \quad (I = 0, 1, \ldots N) \tag{22}$$

This expression holds $(N + 1)$ collocation points. The differential matrix $\mathbf{D}$ is the basic theme fundamental to this scheme. Pursuing a differential matrix involves further mapping into a vector function $H(= [G(\Gamma_0), \ldots, G(\Gamma_N)]^T)$. The collocation points are specified as

$$H' = \sum_{K=0}^{N} \mathbf{D}_{Ki}G(\Gamma_K) = \mathbf{D}H \tag{23}$$

The function $G(\Gamma)$ for the $q$th order derivatives is described as

$$G^q(\Gamma) = \mathbf{D}^q H \tag{24}$$

The matrix **D** is computable, utilizing a similar strategy conferred by Bhatti et al. [37–39]. Now, the spectral collocation method is employed on linearized Equations (19) and (20) to arrive at

$$\mathbf{B}_{i-1}H_i = \mathbf{R}_{i-1} \tag{25}$$

$$G_i(\Gamma_N) = 0, \ \sum_{K=0}^{N} \mathbf{D}_{NK}G_i(\Gamma_K) = 0, \ \sum_{K=0}^{N} \mathbf{D}_{0K}G_i(\Gamma_K) = 0, \ \sum_{K=0}^{N} \mathbf{D}_{0K}^2 G_i(\Gamma_K) = 0 \tag{26}$$

Additionally, we get

$$\mathbf{B}_{i-1} = \mathbf{D}^3 + C_{0,i-1}\mathbf{D}^3 + C_{1,i-1}\mathbf{D}^2 + C_{2,i-1}\mathbf{D} - (M+K)\mathbf{D} + C_{3,i-1} + M\alpha \tag{27}$$

where $b_{\mathbb{S},i-1}(\mathbb{S} = 0, 1, \ldots 3)$ are $(N+1) \times (N+1)$ diagonal matrices along the main diagonal $b_{\mathbb{S},i-1}(\Gamma_N)$. Therefore, we get

$$H_i = G_i(\Gamma_I), \ \mathbf{R}_{i-1} = r_i(\Gamma_I). \ (I = 0, 1, 2, 3, \ldots N) \tag{28}$$

The solutions for $g_i$ are attained using Equations (25) and (26), and Equation (9) becomes linearized. Thus, the Chebyshev pseudo-spectral method is applied in a straightforward fashion to get

$$\mathbb{B} = \mathbb{H}^{-1}\mathbb{S} \tag{29}$$

$$\phi(\Gamma_N) = 1, \ \phi(\Gamma_0) = 0 \tag{30}$$

$$\mathbb{B} = \mathbf{D}^2 + S_c g \mathbf{D} - S_c K_c \tag{31}$$

where $\mathbb{H} = \phi(\Gamma_I)$. The vectors of zeros are defined by $\mathbb{S}$, and Equation (31) is further transformed into the diagonal matrices. Equation (30) is employed over the foremost and last row of $\mathbb{B}$ and $\mathbb{S}$, subsequently.

## 4. Numerical Results and Consultation

This segment premeditates the approximated outcomes for the overall parameters held within the governing equations. Matlab software resorted to exploring the anomalies for the overall effectiveness of the prominent parametric quantities numerically. Figures 2–16 show the prominent parametric quantities of the flow profiles, subsequently. Table 1 portrays the comparative outcomes for $g''(0)$ and $-\phi'(0)$, along with the preceding inquiries across $S_c$ and $K_c$ and the previously published study, while adjusting the prevailing parameters of the governing equations. In Figures 2 and 3, the variation in M for the velocity and concentration distribution is portrayed, and it was noted that the velocity remarkably slowed down and fell with the boundary layer thickness while enhancing the concentration magnitudes by enlarging the values in M. This signifies that the transversal magnetic field resisted the transport phenomenon when an increase resulted in an enhancement of the Lorentz force, which opposed the transport process. An increase in M produced an increase in the Lorentz force due to the interaction of electric and magnetic fields in the electrically conductive fluid. A stronger Lorentz force such as this yielded more resistance to transport. The larger the value of M, the greater the diminution in the hydrodynamic boundary layer thickness. It was also recorded that the variation of M was quietly contrasting to that of $g'$. Through Figures 4 and 5, the alteration in $\beta_1$ for the velocity and concentration distribution is portrayed, and it was recorded that the velocity notably slowed down and the boundary layer thickness increased, although it enhanced the concentration magnitudes by enlarging the values in $\beta_1$. The concentration boundary layer thickness was increasingly thicker, and the concentration transmission slowed down by enlarging $\beta_1$. The figures also depict that more time will be required to induce the concentration boundary layer by enhancing the values in $\beta_1$; in other words, the delayed response seemed to be in concentration transport. In Figures 6 and 7, the variation in $\alpha$ for the velocity and concentration distribution is portrayed, and it is shown that the velocity field remarkably boosted and enhanced the boundary layer thickness while slowing down the concentration

magnitudes by raising the values in $\alpha$. In Figures 8 and 9, the distant numeric value in $K$ for the velocity and concentration distribution is portrayed, and it was found that the velocity remarkably slowed down while enhancing the concentration magnitudes by adding the values in $K$. We found out through Figures 10 and 11 that the variation in $S$ exceptionally decelerated the velocity and concentration distribution, but enhanced the boundary layer thickness for the raising amounts found in $S$. Larger suction generated stronger adherence of the boundary layer to the wall over the stretching sheet regime. This caused an increase in the momentum and boundary layer thickness and decelerated the flow.

Figures 12–14 annotate the influence of $S_c$ and $K_c$ on species concentration successively and show both the recorded parametric quantities slowing down the concentration and the decelerating concentration of the boundary layer thickness. Physically, an increase in the value of $S_c$ reduces the molecular diffusivity that causes a decrease in the thickness of the concentration's boundary layer. Therefore, for the higher values of $S_c$, the concentration of chemically reactive species was bigger and lower for smaller values of the Schmidt number $S_c$. The chemical reaction parameter $K_c$ had a remarkable influence on the concentration, to the point that the chemical reaction benefitted the interface mass transfer. The species concentration fell when $K_c$ reached large values as the destructive chemicals receded. By raising numeric value in $K_c$, the concentration magnitudes were recorded to increase for ($K_c < 0$), and decelerate for ($K_c > 0$). It is worth pointing out that the variation noticed for ($K_c < 0$) was huge compared with the variation for ($K_c > 0$). It can be noticed that the rate of mass transfer was enhanced greatly as the values of the chemical reaction rate rose. Additionally, the rate of mass transfer enhanced when the volume fraction rose. For Figures 15 and 16, it is observed that by taking high values of $\beta_1$ across $K$, the skin friction coefficient deescalates, whereas the Sherwood number escalates.

In the current study, the successive linearization method was employed as it was easy to apply and did not contain rigorous mathematical manipulations, as in the case of some other numerical schemes. All functions of the governing equations were replaced by a power series function expression, which was further expanded and neglected its higher powers. The linearized systems were attained after a slight amount of work to make it useful for solving highly non-linear differential equations. Increasing the number of collocation points $N$ and the number of iterations improved the accuracy of the method. It is also noted that the solutions converged rapidly after putting $N = 60$, $N = 65$, $N = 70$, and so on. The accuracy of the obtained solutions was resolved by comparing them to the previously published results in the literature. The successive linearization method (SLM) converged after 4–5 iterations. The method demonstrated enormous accuracy when compared to other methods adopted in the literature.

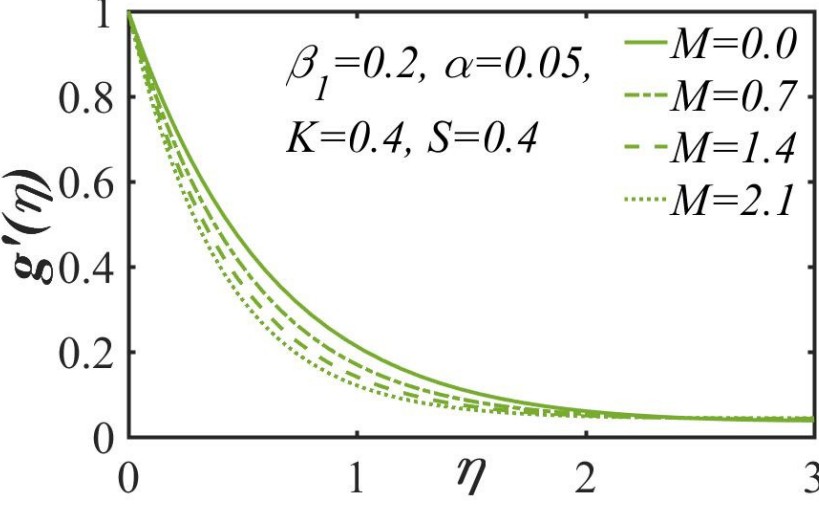

**Figure 2.** Velocity magnitude for $M$.

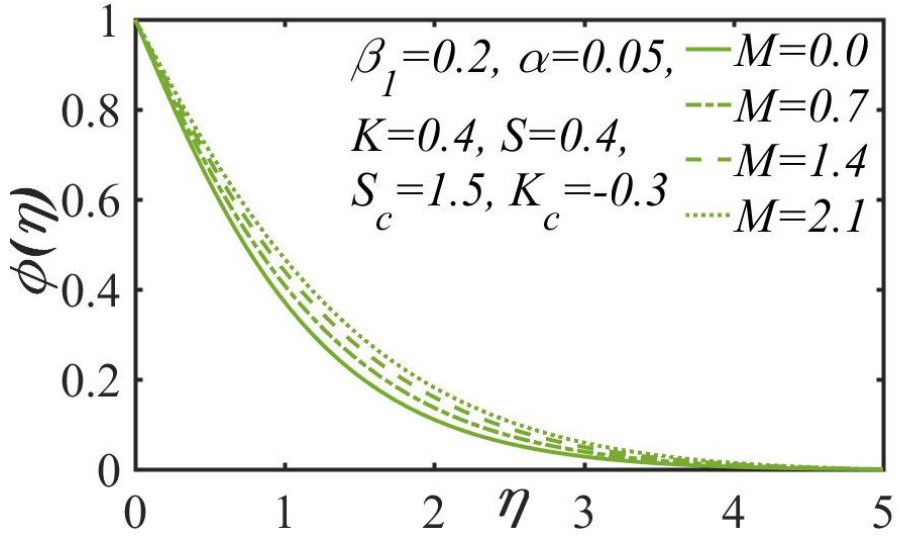

**Figure 3.** Concentration magnitude for *M*.

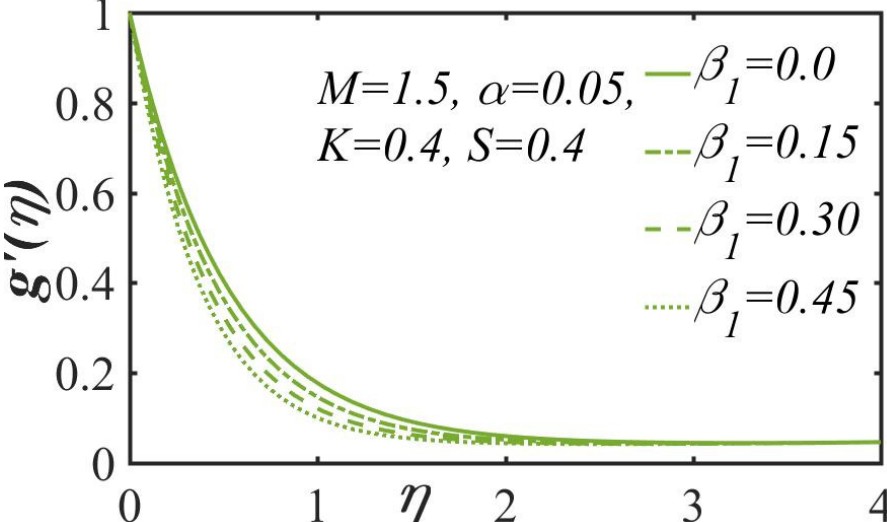

**Figure 4.** Velocity magnitude for $\beta_1$.

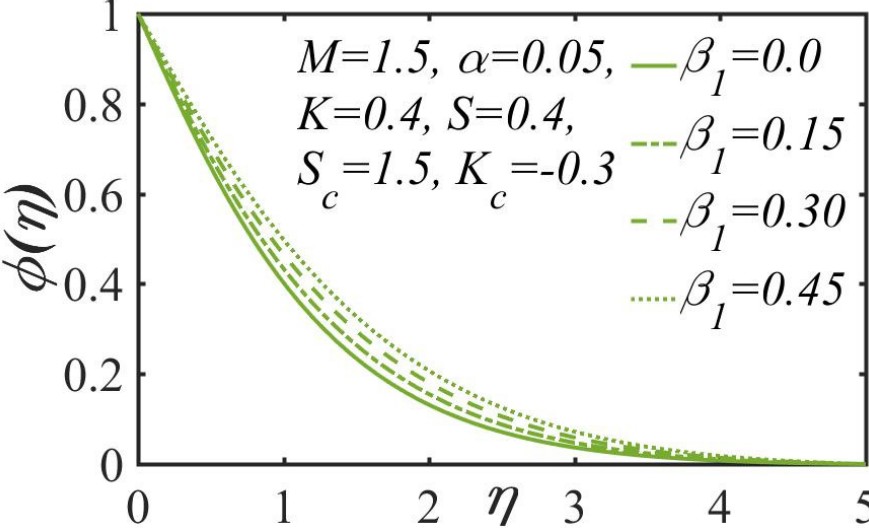

**Figure 5.** Concentration magnitude for $\beta_1$.

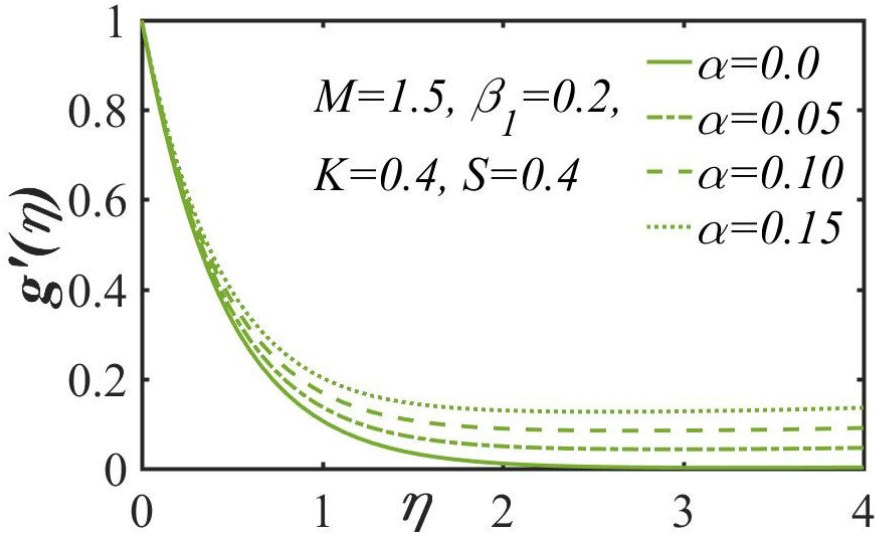

**Figure 6.** Velocity magnitude for *α*.

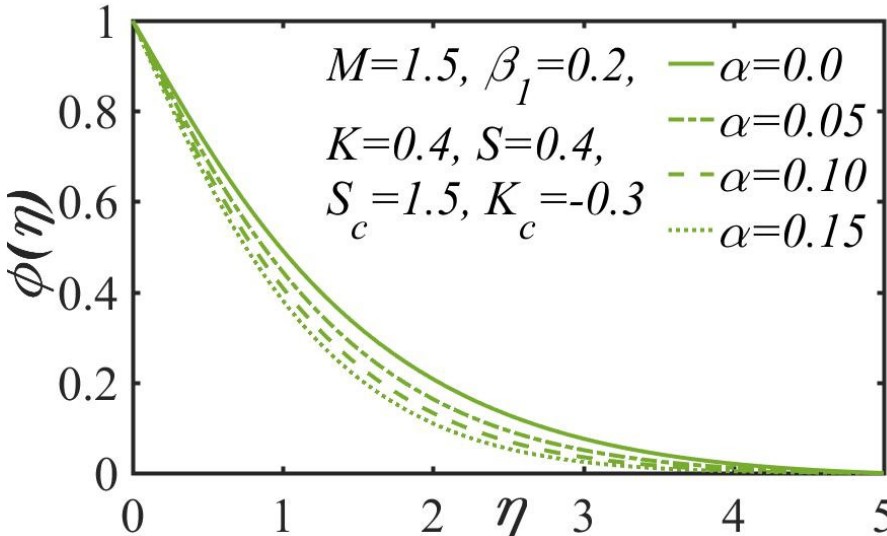

**Figure 7.** Concentration magnitude for *α*.

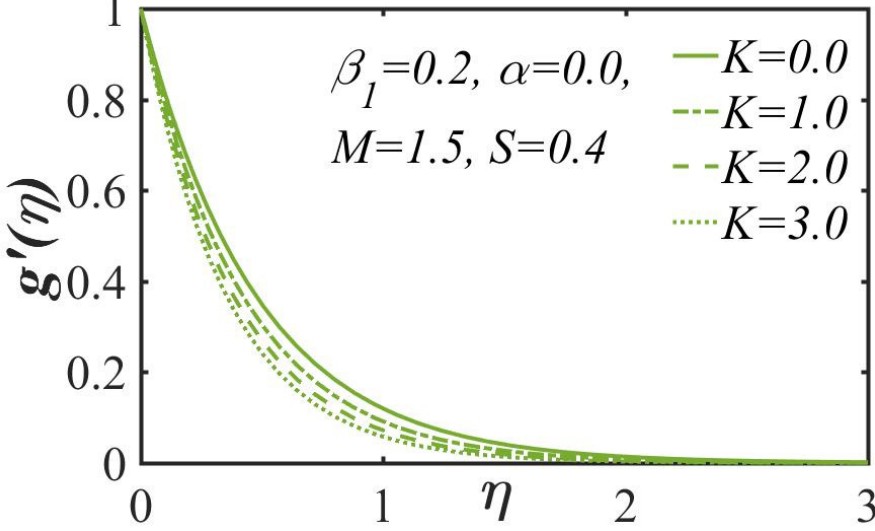

**Figure 8.** Velocity magnitude for *K*.

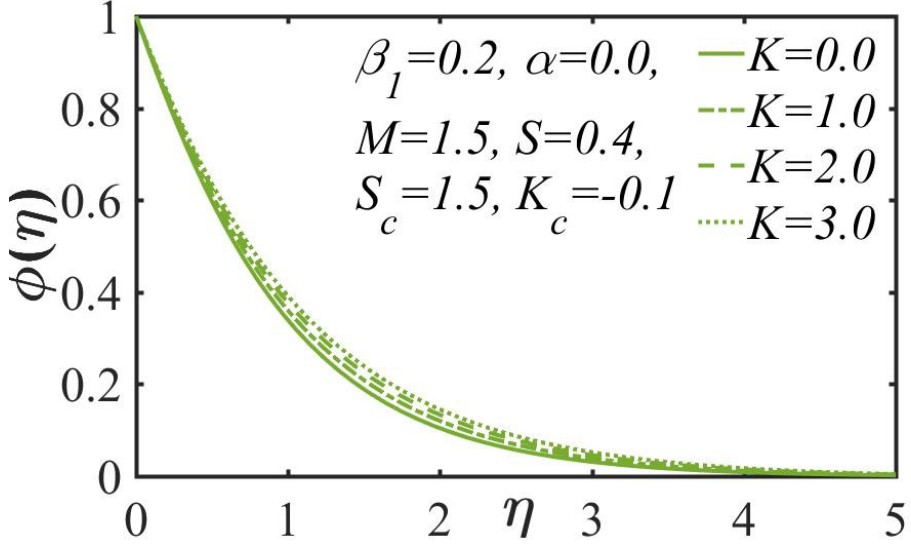

**Figure 9.** Concentration magnitude for *K*.

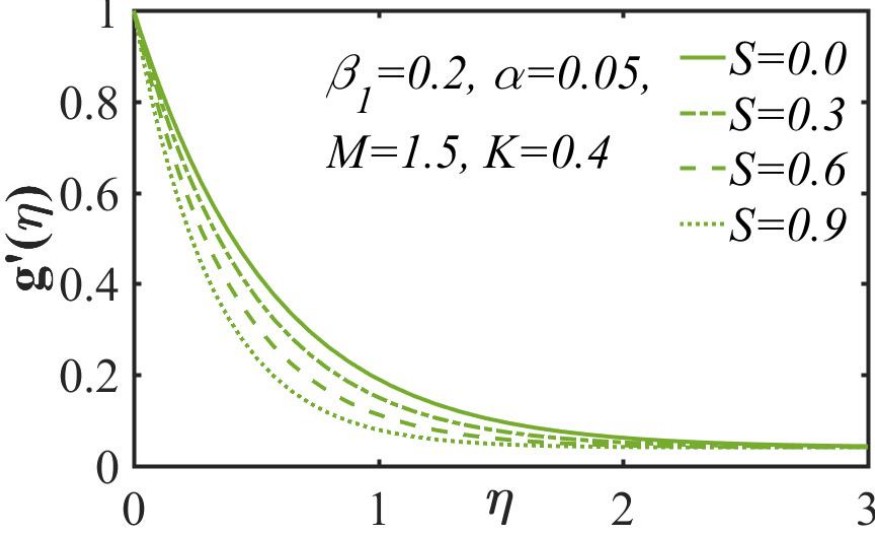

**Figure 10.** Velocity magnitude for *S*.

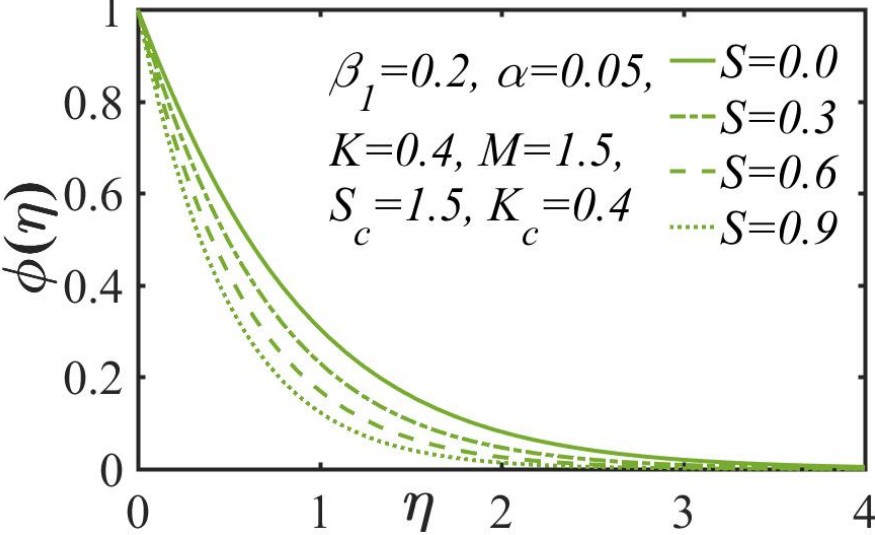

**Figure 11.** Concentration magnitude for *S*.

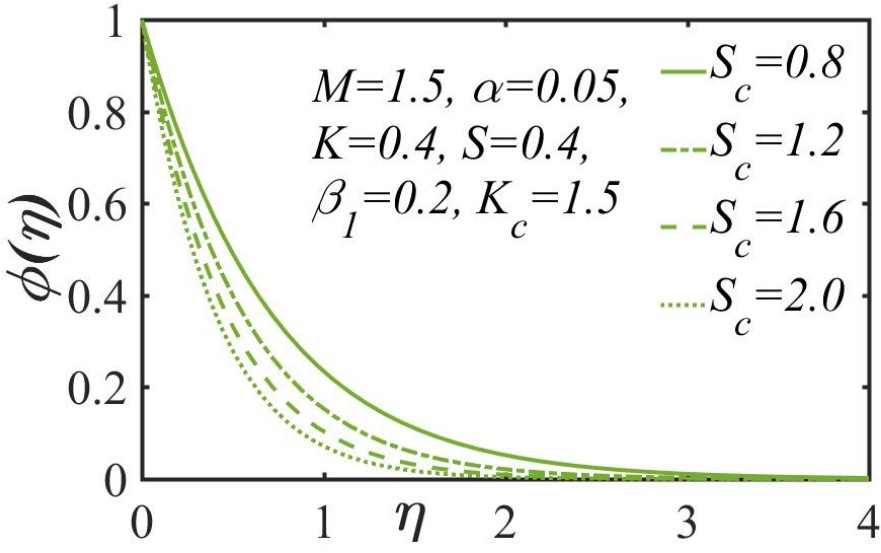

**Figure 12.** Concentration magnitude for $S_c$.

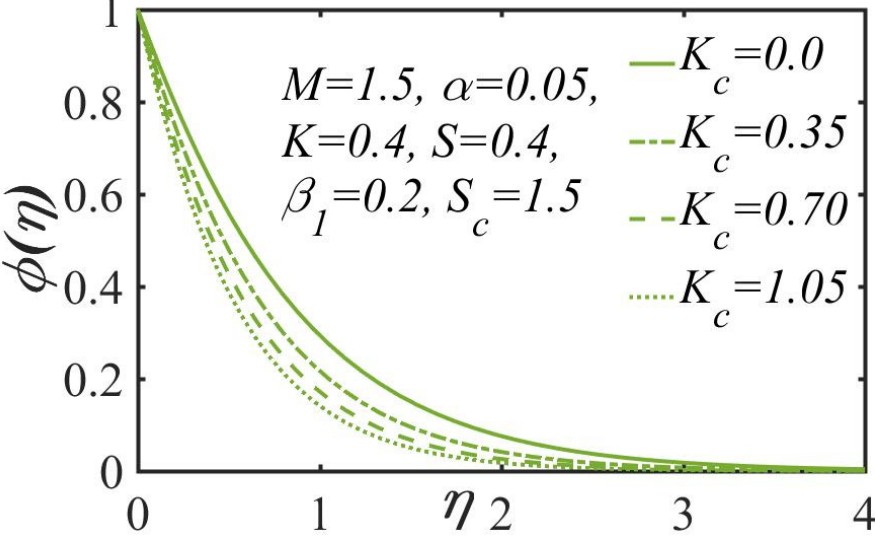

**Figure 13.** Concentration magnitude for $K_c \geq 0$.

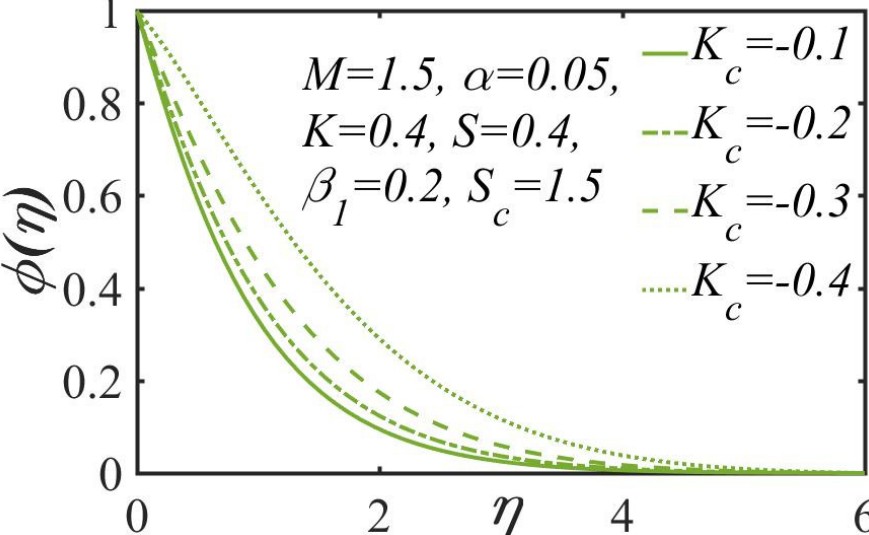

**Figure 14.** Concentration magnitude for $K_c < 0$.

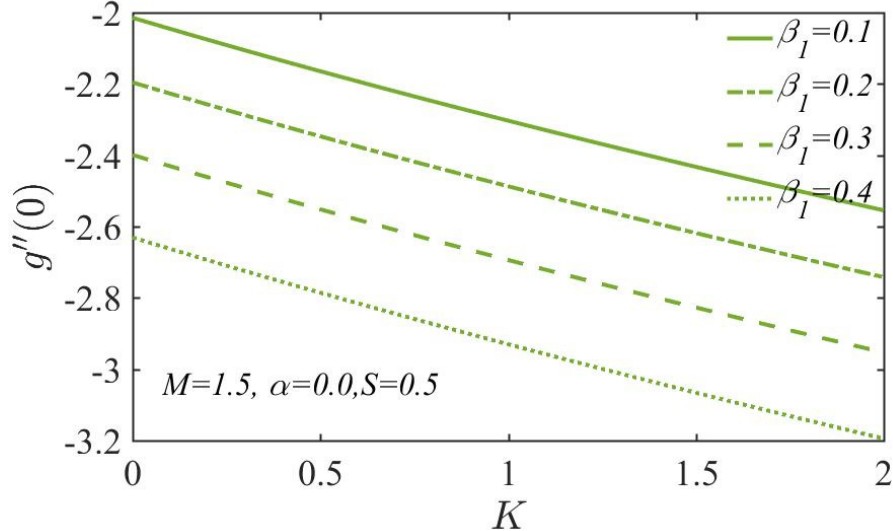

**Figure 15.** Skin friction coefficient with the fluctuation of $\beta_1$.

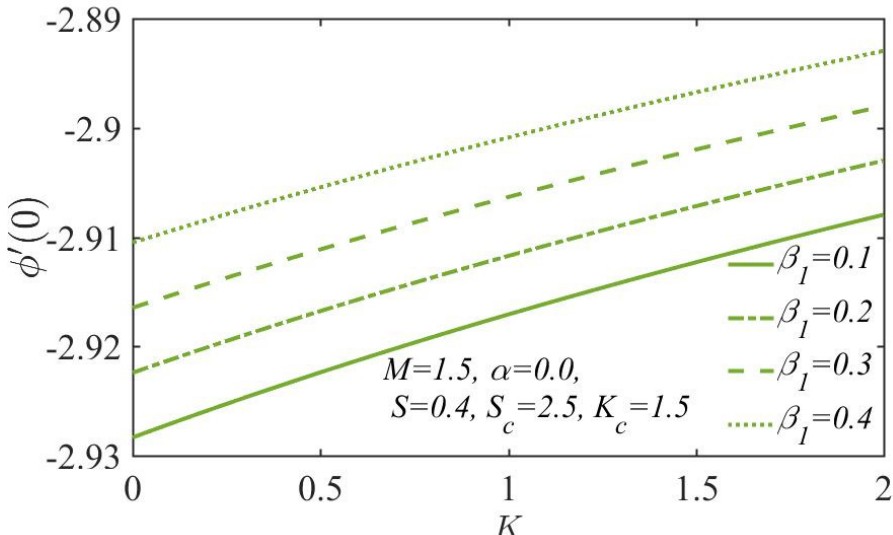

**Figure 16.** Sherwood number with the variation of $\beta_1$.

**Table 1.** Comparability for the current outcomes of $g''(0)$ and $-\phi'(0)$, along with the preceding inquiries across $S_c$ and $K_c$ by setting the values $M = 1$, $\beta_1 = \alpha = 0.2$, and $K = S = 0$.

| $S_c$ | $K_c$ | Current Results for $g''(0)$ | [48] | Current Results for $-\phi'(0)$ | [48] |
|-------|-------|------------------------------|------|----------------------------------|------|
| 1.0   | 1.0   | 1.272470                     | 1.272469 | 1.167862                     | 1.16786 |
| 1.2   | 1.0   |                              |      | 1.284681                         | 1.28467 |
| 1.5   | 1.0   |                              |      | 1.443482                         | 1.44347 |
| 1.0   | 1.2   |                              |      | 1.252273                         | 1.25226 |
| 1.0   | 1.5   |                              |      | 1.368854                         | 1.36885 |

## 5. Conclusions

The current analysis demonstrates the impact of mass transfer into MHD upper-convected Maxwell (UCM) fluid flow on a stretched, permeable plate in the vicinity of the stagnation point. The impacts of porosity and suction were also investigated. The governing partial differential equations for the flow problem were reformed into ordinary differential equations by using similarity transformations. The numerical outcomes for the uprising non-linear boundary value problem were determined by implementing the successive linearization method (SLM), which utilizes both Chebyshev interpolating

polynomials and Gauss–Lobatto collocation points via Matlab software. The approximated results were sketched as graphs and compared with the preceding investigated literature. The succeeding attentions are recorded as follows:

- The variation in M for the velocity distribution remarkably slowed down while enhancing the concentration magnitude.
- The variation in $\beta_1$ decelerated the velocity profile, although it enhanced the concentration magnitudes.
- The variation in $\alpha$ notably boosted the velocity magnitudes while it slowed down the concentration magnitudes.
- Raising the numeric value in *K* turned out to decelerate the distribution while enhancing the concentration magnitudes.
- By enlarging *S*, there were significant slowdowns in the velocity and concentration distributions.
- The concentration magnitudes successively decelerated across both parametric quantities $S_c$ and $K_c$.
- The concentration field had contradictory conduct across ($K_c > 0$) and ($K_c < 0$).
- The skin friction coefficient decelerated, whereas the Sherwood number enhanced across the variation in $\beta_1$.

**Funding:** This research received no external funding.

**Conflicts of Interest:** The author declare no conflict of interest.

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
