# Peer review of "The Effectiveness of Mass Transfer in the MHD Upper-Convected Maxwell Fluid Flow on a Stretched Porous Sheet near Stagnation Point: A Numerical Investigation"

_inventions, doi:10.3390/inventions5040064_

Round 1
Reviewer 1 Report
This investigation concerns a magnetohydrodynamic (MHD) upper-convected Maxwell fluid flow subject to porous medium and mass transfer. Suitable similarity transformations have been employed for converting the governing nonlinear differential equation system into a system of ordinary differential equations. Successive linearization method along with Chebyshev spectral collocation technique were used to solve the final dimensionless differential equations. Solutions for velocity and concentrations profiles were plotted for various dimensionless parameters relevant to the problem. The whole paper is interesting and it addresses the problem of an MHD non-Newtonian fluid flow subject to porous medium and mass transfer, which can have great industrial interest. However, some corrections/improvements are needed:
- The grammar and the syntax of the paper need some corrections. For example, at the first line of the introduction section, someone can read “regarding to non-Newtonian fluids”. The correct sentence is “regarding non-Newtonian fluids” or “with regards to non-Newtonian fluids”. It is recommended that the whole study is checked by a native English speaker.
- A schematic representation of the flow should be included.
- Although a number of relevant studies are cited, some others equally interesting could be included, for example:
- Influence of Ramped Wall Temperature and Ramped Wall Velocity on Unsteady Magnetohydrodynamic Convective Maxwell Fluid Flow. doi: 3390/sym12030392. This is a very interesting new study regarding an MHD Maxwell fluid flow.
- Peristaltic Propulsion of Jeffrey Nanofluid with Thermal Radiation and Chemical Reaction Effects. doi: 3390/inventions4040068. This a relatively new study with regards to an MHD non-Newtonian nanofluid with chemical reaction.
- Effect of radiation and Navier slip boundary of Walters’ liquid B flow over a stretching sheet in a porous media. doi: 1016/j.ijheatmasstransfer.2018.02.084. This is an interesting study regarding the effect of magnetohydrodynamics and porous media on a non-Newtonian (Walters’ liquid B) flow which is also utilizing similarity transformation.
- Impact of Thermal Radiation and Heat Source/Sink on MHD Time-Dependent Thin-Film Flow of Oldroyed-B, Maxwell, and Jeffry Fluids over a Stretching Surface. doi: org/10.3390/pr7040191. This study considers a number of MHD non-Newtonian fluids over a linear stretching surface.
- Micromagnetorotation of MHD Micropolar Flows. doi:10.3390/sym12010148. This is an interesting new study regarding the influence of the magnetic field on a non-Newtonian (micropolar) fluid flow.
- More discussion about the physical meaning of the solutions should be included (e.g. how Lorentz forces affect the flow when the magnetic parameter M changes).
- Reference 13 should be checked, there is a miswriting (i.e. [13] in the beginning of the text).
Author Response
Please, see the attachment Ms word file.

Reviewer 2 Report
inventions-943921-
The present inquiry piles the influence of mass transfer in MHD, upper-convected Maxwell (UCM) fluid flow on a stretchable porous subsurface. The governing partial differential equations for the flow problem are reformed to ordinary differential equations by operating similarity transformations
- A lot of work on Maxwell fluid is already published, what is new and what is different in your work?
- What is the objectives, methodology, briefly point out the main results they obtained in the abstract
- The introduction section should be improved avoiding lump sum references such as XXXXX [6-9], OR 1, 2, 3, 4, 5; all references should be cited with detailed and specific descriptions. Include Novelty. The Reference need to be updated. The authors should refer to the following recent articles.
https://doi.org/10.3390/coatings10100998
https://doi.org/10.1016/j.icheatmasstransfer.2020.105039.
https://doi.org/10.3390/e22040480
https://doi.org/10.1016/j.csite.2020.100732.
- How it is possible to include mass transfer without heat transfer includes details.
- Write more details for equations 2 and 3 remove typo mistakes
- Why such boundary conditions used?
- Problem modelling and mathematical sections should be improved, the authors have listed all equation but without any comments and description.
- Support the basic model equations with strong references
- Skin friction derivation is wrong revise and correct it (eq 13)
- Why parameter is taken in such range?
- Why such method is used? Include details. what about the convergence.
- The authors should do a better job on commenting the results. A reasonable physical explanation should be provided for the observed trends, not only report what is graphically seen in the Figures;
- You can compare your results with previous work.
- Please consider improving the manuscript quality and scientific soundness, the manuscript cannot be accepted for publication it is not clear to the reader the novelty of this paper.
- Revise conclusion and make it precious
Author Response
Please see the attached Ms word file.

Round 2
Reviewer 1 Report
All the revisions suggested have been included in the manuscript. The current version of the paper is of high quality. It is recommended it for publication, good luck!
Reviewer 2 Report
The manuscript is improved and the author adressed all my comments, I accept the manuscript now.